# A Novel Recombinant Human FGF21 Analog with High Glycosylation Has a Prolonged Half-Life and Affects Glycemic and Body Weight Control

**DOI:** 10.3390/ijms26062672

**Published:** 2025-03-16

**Authors:** Pei Du, Ting Wang, Rong Wang, Shang Liu, Hang Wang, Hongping Yin

**Affiliations:** 1School of Life Science and Technology, China Pharmaceutical University, 639 Long Mian Avenue, Nanjing 211198, China; 1520220090@cpu.edu.cn (P.D.); 1821030666@stu.cpu.edu.cn (R.W.); 1520220083@cpu.edu.cn (S.L.); hangw@cpu.edu.cn (H.W.); 2Glycomics and Glycan Bioengineering Research Center (GGBRC), College of Food Science and Technology, Nanjing Agricultural University, 1 Weigang, Nanjing 210095, China; wangting@njau.edu.cn

**Keywords:** FGF21, recombinant fusion protein, mucin fusion, adipocyte, glycemic control, obesity

## Abstract

Fibroblast growth factor 21 (FGF21), a hormone-like protein, plays a crucial role in enhancing glucose and lipid metabolism, offering promising therapeutic avenues for conditions such as nonalcoholic steatohepatitis and severe hypertriglyceridemia. Despite its potential, this protein’s limited stability and brief half-life pose significant challenges for its use in clinical settings. In this study, we created an FGF21 analog (named FGF21-164) that is a mutant of FGF21 and fused it with the tandem repeat sequence of human CD164. FGF21-164, characterized by extensive glycosylation and sialylation, exhibits enhanced pharmacokinetic properties, particularly in terms of its significantly longer half-life compared to its native form. The in vitro efficacy of FGF21-164 was evaluated using 3T3-L1-induced adipocytes. The protein demonstrated a dose-dependent increase in glucose uptake and effectively decreased lipid droplet accumulation surrounding the adipocytes. The in vivo activity of FGF21-164 was evaluated in *leptin-deficient* (*ob*/*ob*) and diet-induced obesity (DIO) mice. A single subcutaneous dose of FGF21-164 led to a rapid decrease in blood glucose levels and sustained normal fasting glucose levels for up to 28 days. Additionally, repeated dosing of FGF21-164 significantly curbed weight gain and reduced hepatic fat accumulation in DIO mice.

## 1. Introductions

Fibroblast growth factor 21 (FGF21), a 19.5 kDa hormone-like protein, is part of the fibroblast growth factor family [1]. FGF21 interacts with fibroblast growth factor receptors (FGFR1c) and β-Klotho, primarily found on adipocytes, facilitating the transmission of signals into the cells. [2]. FGF21 plays a role in regulating lipid [3] and glucose metabolism [4] and energy expenditure [5]. Consequently, FGF21 has been explored as a potential treatment for type 2 diabetes [6]. Nevertheless, its therapeutic application is hampered by two major obstacles. One significant limitation is the short half-life of native FGF21 (approximately 30 min) [7,8], partly attributed to renal clearance due to its moderate molecular weight but primarily to rapid inactivation. The N-terminal amino acid of native FGF21 protein is His-Pro-Ile-Pro (HPIP), which is prone to proteolytic cleavage, while the C-terminal region of native FGF21 is vulnerable to degradation by fibroblast-activating protein (FAP), an endopeptidase that removes the terminal 10 amino acids of native FGF21, significantly impairing its binding affinity to the coreceptor β-Klotho [9,10]. Significantly, FAP exhibits elevated expression levels in the livers of individuals with type 2 diabetes or metabolic-related conditions. [6,11]. In addition, the proline at position 171 and alanine at position 180 of FGF21 are susceptible to cleavage by site-specific proteases [12,13]. To enhance the therapeutic potential of FGF21, strategies such as mutagenesis for improving stability, fusion with a carrier, and PEGylation for extending general circulation have been explored [14,15].

Developing FGF21 analogs with extended plasma half-lives is essential for exploring its therapeutic potential. In earlier research, we employed a tandem repeat (TR) amino acid sequence from human CD164 (hCD164, also named MUC24) to enhance the half-life of myeloid-derived growth factor [16]. The TR fusion carrier (Ser110-Asp162 of hCD164) is heavily glycosylated, featuring 2 N-glycosylation sites and 23 O-glycosylation sites. As a result, the TR fusion protein exhibited extensive glycosylation, with sialic acid modifications enriching the glycan termini. Since glycosylation plays a critical role in protein stability, this TR fusion carrier could serve as an optimal fusion partner that can prolong the plasma half-life of FGF21.

Here, we describe a novel human FGF21 analog, designated as FGF21-164, which was generated through fusion protein technology and co-encoded by a mutated FGF21 gene and the TR gene of hCD164. FGF21-164 was expressed via a eukaryotic cell, and its physicochemical and pharmacokinetic properties were thoroughly evaluated. Subsequently, the biological activity of FGF21-164 was assessed at the cellular level. Furthermore, FGF21-164 was tested in *ob*/*ob* and diet-induced obese (DIO) mice to demonstrate its activity.

## 2. Results

### 2.1. Construction and Biophysical Characterization of FGF21-164

We screened approximately 450 clones and obtained seven subcloned cell lines with high FGF21-164 expression (Figure 1B). According to the protein expression efficiency, clone F3 was finally selected for subsequent experiments. The purified FGF21-164 exhibited a prominent band ranging from 48 to 63 kDa on gel, and it was confirmed as FGF21-164 via immunoblotting via an anti-FGF21 antibody (Figure 1C). Although the protein FGF21 was detectable in the transfected cells, it exhibited two closely migrating bands via immunoblot analysis (Figure 1B). Upon purification, we obtained FGF21 protein that migrated as a single band under denaturing conditions; however, subsequent immunoblotting of the purified protein still revealed two adjacent bands (Figure 1C). SEC-HPLC was employed to assess the hydrodynamic properties and purity of FGF21-164 in physiological solution, revealing a molecular weight of 85.65 kDa and a purity exceeding 95.92% (Figure 1D). Further validation of the molecular weight was performed using MALDI-TOF MS, which displayed values of approximately 42751.5 [M + H] ^+^ and 21601.173 [M + 2H] ^2+^ (Figure 1E). The secondary structures of FGF21 and FGF21-164 were evaluated using Circular Dichroism (CD), and the corresponding waveforms were approximately similar in wavelength from 200 to 260 nm (Figure 1F). This result suggested that the composition of the secondary structure derived from CD spectra showed minor differences between FGF21-164 and FGF21, indicating that the TR fusion had minimal impact on the structural integrity of FGF21.

### 2.2. High Glycosylation of FGF21-164

To verify glycosylation and analyze its specific patterns, we conducted N- and O-glycan profiling on FGF21-164. A total of 39 distinct N-glycans were detected in FGF21-164, including 11 types of biantennary structures, 22 types of tri-antennary structures, and six types of tetra-antennary structures (Figure 2A). Additionally, the O-glycans of FGF21-164 were analyzed following β-elimination-mediated chemical release, employing the established analytical workflow. The investigation identified ten distinct O-glycosylated structures categorized as Core 2, 3, or 4 subtypes (Figure 2B). Notably, terminal Neu5Ac modifications were predominantly observed in both N-linked and O-linked glycans in comprehensive composition analysis.

### 2.3. Identification of the In Vitro Bioactivity of FGF21-164

Adipocyte differentiation in 3T3-L1 cells was assessed through lipid droplet accumulation. When cultured under established differentiation conditions (Figure 3A), bright-field microscopy revealed readily detectable intracellular lipid deposits by Day 10 of the induction process. The in vitro bioactivity of FGF21-164 was evaluated through glucose uptake measurements. Experimental data revealed dose-responsive enhancement of glucose assimilation in differentiated 3T3-L1 adipocytes following treatment with FGF21-164 (Figure 3B). To determine whether FGF21-164 binds to FGFR and downstream signaling transduction, we examined the Erk1/2 activation status in 3T3-L1 adipocytes. Immunoblotting demonstrated a time-dependent augmentation of Erk1/2 phosphorylation during the initial 30 min exposure to FGF21-164, followed by progressive signal attenuation (Figure 3C). As shown in Figure 3D, the cells in the induction medium (Model) formed many lipid droplets in 3T3-L1 adipocytes, whereas no lipid droplets formed in the undifferentiated cells (Ctrl). However, compared with the model cells, the adipocytes treated with FGF21-164 for 6 days presented fewer lipid droplets (Figure 3D). The lipid-lowering effect of FGF21-164 was further validated through quantitative analysis, as illustrated in Figure 3D.

### 2.4. Pharmacokinetic Analysis of FGF21-164

Quantitative LC-MS/MS bioanalysis of C57BL/6 mice revealed the following kinetic parameters for FGF21-164: elimination phase—terminal half-life (t_1_/_2_) = 2.60 ± 0.335 h; exposure metrics—peak plasma concentration (C_max_) = 504 ± 49.9 µg·mL^−1^; total systemic exposure (AUC_0–t_) = 949 ± 46.4 h·µg·mL^−1^; extrapolated AUC_0–_∞ = 1040 ± 55.6 h·µg·mL^−1^; volume of distribution (Vd_ss_) = 64.2 ± 1.78 mL·kg^−1^; and plasma clearance rate = 21.9 ± 1.22 mL·h^−1^·kg^−1^. The plasma concentration–time profile (Figure 4) demonstrated dose-proportional kinetics with a rapid distribution followed by first-order elimination.

### 2.5. Effect of Administering a Single Dose of FGF21-164 on ob/ob Mice

Leveraging FGF21’s established mechanism of targeting adipocyte β-Klotho/FGFR1c complexes for insulin-independent glycemic regulation, we conducted pharmacodynamic validation of the FGF21-164 analog in leptin-deficient (*ob*/*ob*) murine models to assess therapeutic equivalence in glucose homeostasis. According to a previous report, *ob*/*ob* mice exhibit basal hyperglycemia due to a lack of the leptin gene. Thus, *ob*/*ob* mice were administered FGF21-164 or PBS subcutaneously to evaluate the hypoglycemic effect. Longitudinal glycemic profiling via ultrasensitive glucometry revealed time-dependent hypoglycemic responses in the *ob*/*ob* mice. As a result, the blood glucose levels were reduced in all the *ob*/*ob* mice after the oral administration of glucose. It was noteworthy that the blood glucose levels in the 6 mg·kg^−1^ FGF21-164 treatment group were significantly lower compared to the model at 4, 7, and 8 h (Figure 5A). In contrast, compared with the model, the blood glucose levels were significantly lower in the FGF21-164-treated groups, especially in the high-dose group (12 mg·kg^−1^) (Figure 5B).

Longitudinal metabolic profiling was conducted to evaluate FGF21-164’s sustained effects on dynamic glucose homeostasis in *ob*/*ob* mice, incorporating fasting plasma glucose measurements and a standardized 1.5 g·kg^−1^ oral glucose challenge test at the 7-, 21-, and 28-day post-dose intervals following a single subcutaneous administration. At 7, 21, and 28 days, the fasting blood glucose levels in the model were slightly lower, but there was no significant difference from those at 0 days (9.875 ± 0.189 mmol·L^−1^), and the fasting blood glucose levels at 7, 21, and 28 days were 10.05 ± 0.902, 8.7 ± 0.876, and 7.725 ± 0.581 mmol·L^−1^ in the model group (Figure 6A,D,G). However, in the FGF21-164 treatment group, fasting blood glucose levels decreased significantly on days 7, 21, and 28. In the 6 mg·kg^−1^ FGF21-164 treatment group, the fasting blood glucose levels at 0, 7, 21, and 28 days were 9.3 ± 0.435, 6.16 ± 0.35, 6.5 ± 0.527, and 5.98 ± 0.361 mmol ·L^−1^ (Figure 6A,D,G). In the 12 mg·kg^−1^ FGF21-164 treatment group, the fasting blood glucose levels at 0, 7, 21, and 28 days were 12.18 ± 1.199, 6.68 ± 0.267, 8.38 ± 0.727, and 6.28 ± 0.585 mmol ·L^−1^ (Figure 6A,D,G). In comparison to the PBS-treated group (the model), the FGF21-164-treated group showed significantly lower blood glucose levels at 30 min following glucose administration (Figure 6B,E,H). Furthermore, the increase in blood glucose AUC was notably attenuated in the FGF21-164 group at both 7 and 21 days after a single administration (Figure 6C,F).

### 2.6. Effect of Repeated FGF21-164 Administration on DIO Mice

Since no weight changes were observed with a single dose of FGF21-164 (Figure 6J), we evaluated the weight changes in the DIO mice after repeated doses of FGF21-164. The DIO mice were administered intraperitoneal injections of 3 mg·kg^−1^ of FGF21-164 every other day. Compared with the age-matched Ctrl mice, the DIO mice in the model had significantly greater body weights and a continuous growth trend, whereas the DIO mice treated with FGF21-164 slowly lost body weight, with 17.31% (6.0 g) weight loss after the 19 days of treatment compared with the DIO mice that were not subjected to FGF21-164 treatment (Figure 7A). The results of a histological examination of the liver sections stained with Oil Red O are presented in Figure 7B. A large number of Oil Red O-stained red lipid droplets accumulated in the livers of the DIO mice, and this accumulation was significantly reduced by the FGF21-164 treatment. The images also reveal that hepatocytes in the DIO mice exhibited marked cellular hypertrophy (characterized by an increased cytoplasmic volume and nuclear enlargement), consistent with lipid accumulation and metabolic stress. In contrast, hepatocytes from the FGF21-164-treated group displayed a smaller cross-sectional area compared to the model, indicative of attenuated steatosis and restored cellular architecture.

## 3. Discussion

In this study, we aimed to develop advanced FGF21-based therapeutics by enhancing this protein’s retention in the blood and stability in order to address challenges relating to clinical use. Using hCD164 fusion technology, an FGF21-164 fusion protein was engineered, and its structural properties, biological activity, and glucose-lowering effects were assessed. The results demonstrated that FGF21-164 had been successfully produced, showing prolonged blood retention, sustained blood glucose regulation, and reduced hepatic fat accumulation.

To generate the FGF21-164 fusion protein, the N-terminal HPIP sequence was removed from FGF21 (ΔHPIP), and cysteine substitutions were introduced at positions 118 and 134 (L118C and A134C) to enhance its stability [13]. Then, the C-terminus of mutant FGF21 was fused with the N-terminus of the TR region of hCD164 at the gene level. FGF21-164 could be expressed in 293F cells, and denaturation and refolding steps were not necessary for purification (Figure 1B). FGF21 could also be expressed in 293F cells, but only two bands corresponding to FGF21 could be recognized by the FGF21 antibody (Figure 1B). The two bands of FGF21 may indicate that the protein was hydrolyzed by site-specific proteases [9]. Following purification, FGF21 migrated as a single band under denaturing SDS-PAGE conditions. However, Western blot analysis of the purified protein consistently revealed two distinct immunoreactive bands (Figure 1B), indicative of structural instability in the FGF21 molecule. In addition, FGF21 was degraded during storage, so it was not used as a control in the subsequent experiments.

The extended serum half-life of FGF21-164 (2.6 h, as shown in Figure 4) can be attributed to synergistic mechanisms involving molecular size, glycosylation patterns, and charge-mediated interactions. SEC-HPLC analysis revealed that FGF21-164 possesses a hydrodynamic radius corresponding to ~85.65 kDa (Figure 1D), significantly larger than its calculated molecular weight (24.34 kDa). This increased molecular size likely reduces glomerular filtration rates by impeding passage through the renal filtration barrier [17]. FGF21-164 was identified as a heavily glycosylated and sialylated protein, characterized by both N- and O-linked glycans terminating in α-2,6/α-2,3-linked Neu5Ac (sialic acid) residues (Figure 2A,B). These sialylation modifications confer FGF21-164 a key advantage via extending its half-life. First, sialylation-mediated masking of galactose residues in FGF21-164 effectively inhibits recognition by the asialoglycoprotein receptor (ASGPR) abundantly expressed on hepatic sinusoidal endothelial cells. The negatively charged carboxyl groups of sialic acids generate steric and electrostatic repulsion, further inhibiting ASGPR binding [18,19,20]. Second, sialylation increases FGF21-164’s hydrophilicity, reducing its aggregation propensity and improving its plasma solubility. Concurrently, the negative surface charge repels similarly charged basement membranes and podocytic epithelia in the glomerulus, synergistically decreasing renal clearance rates [17,21]. Additionally, sialic acid capping may mitigate immune recognition of FGF21-164, thereby reducing phagocytic uptake and proteolytic degradation [22]. Collectively, these structural and chemical modifications synergistically prolong FGF21-164’s circulation time while maintaining functional integrity.

The functional efficacy of FGF21-164 was examined in 3T3-L1 adipocytes [23,24]. Mirroring native FGF21, FGF21-164 demonstrated robust activation of FGF21R signaling pathways, evidenced by its ability to induce the phosphorylation of Erk1/2 [25], a critical downstream effector in FGF21-mediated signaling cascades (Figure 3C). Subsequently, the biological activity of FGF21-164 was assessed through a glucose uptake assay. The ability of adipocytes to absorb glucose increases with an increasing FGF21-164 concentration (Figure 3B). Moreover, the CCK-8 assay results revealed that FGF21-164 did not affect 3T3-L1 cell growth. Oil Red O staining revealed that the presence of 100 µg·mL^−1^ FGF21-164 effectively reduced the accumulation of ring-shaped lipid droplets around adipocytes, and quantitative testing confirmed this finding (Figure 3D).

The in vivo activity of FGF21-164 was evaluated in *ob*/*ob* mice and DIO mice [26,27]. Notably, although the average terminal *t_1/2_* of FGF21-164 was only 2.6 h, its efficacy was maintained for almost 28 days. The fasting blood glucose levels of *ob*/*ob* mice remained at normal levels (5.98–6.28 mol·L^−1^) after 28 days of single administration, whereas the blood glucose levels of the nontreated *ob*/*ob* mice were as high as 7.73 mol·L^−1^ (Figure 6G). This glucose-maintaining effect is superior to that of other FGF21 fusion proteins that have been reported. For example, a single subcutaneous injection of Fc-FGF21[R19V][N171], also called PF-06645849, affected blood glucose in *ob*/*ob* mice for only 14 days [28]. A single subcutaneous injection of ELP-FGF21 affected blood glucose in *ob*/*ob* mice for only 5 days [29]. A single subcutaneous injection of Fc-FGF21 variants affected blood glucose in *db/db* mice for only 9 days [30]. However, the effect of a single injection of FGF21-164 over the course of 28 days still requires further pharmacodynamic studies.

There are numerous reports that while the efficacy of FGF21 and its analogs/mimetics for glycemic control is rather mild and inconsistent [31,32], it does have an effect on weight loss. Our experimental investigations revealed that FGF21-164 significantly suppressed body weight gain and attenuated hepatic lipid accumulation in diet-induced obese (DIO) mice, as demonstrated in Figure 7. Some investigations have demonstrated that FGF21 and its analogs target multiple stages in the pathogenic cascade of metabolic-dysfunction-associated steatohepatitis (MASH). These effects include the clearance of lipids from hepatocytes, liver cell stress protection, the inhibition of apoptosis and proinflammatory signaling pathways, and the suppression of the differentiation of hepatic stellate cells into collagen-producing myofibroblasts [33]. Clinical studies have suggested that a recombinant FGF21 analog (pegozafermin) can be used to treat nonalcoholic steatohepatitis (NASH) and severe hypertriglyceridemia [34,35]. Clinical trial data indicated that pegozafermin treatment potentially enhances multiple metabolic parameters, e.g., reducing hepatic steatosis, improving inflammatory and fibrotic biomarkers, and allowing better regulation of lipid profiles and glucose metabolism [36]. Liver histology improvements were demonstrated in an open-label cohort of biopsy-proven NASH patients treated with pegozafermin [37]. A phase 2b trial suggested that patients with NASH treated with pegozafermin experienced improvements in fibrosis [38]. The efficacy of using FGF21-164 to reduce hepatic fat accumulation in DIO mice suggested that it may help treat non-alcoholic fatty liver disease (NAFLD). However, the current study primarily focuses on the effects of FGF21-164 on body weight regulation and hepatic lipid accumulation in diet-induced obese (DIO) mice. Advanced stages of NAFLD/NASH, characterized by histopathological features such as ballooning degeneration and fibrosis, may necessitate longer-term studies or alternative preclinical models (e.g., STAM mice, methionine-choline-deficient-diet-fed rodents, or humanized liver chimeric models) to fully elucidate the therapeutic potential of FGF21-164 in late-stage disease contexts. Furthermore, the systemic metabolic effects of FGF21-164—particularly its interplay with extrahepatic tissues (e.g., adipose and skeletal muscle) and long-term metabolic homeostasis—require comprehensive investigation. To address this, multi-tiered transcriptomic approaches can be adopted to provide unprecedented resolution in dissecting FGF21-164’s actions within the complex tissue microenvironment, and systems-level transcriptomic approaches are essential to decode its pleiotropic metabolic effects and accelerate clinical translation.

While this study provides critical insights into the therapeutic potential of FGF21-164, several limitations must be acknowledged to contextualize its findings. First, our reliance on murine models (*ob*/*ob* and DIO mice) inherently restricts the direct translation of the results to human metabolic pathophysiology. Murine systems, though widely used in obesity and diabetes research, exhibit fundamental differences in lipid handling, insulin sensitivity, and immune responses compared to humans [39]. For instance, the absence of certain FGF21-regulated pathways in mice (e.g., bile acid metabolism divergence) may lead to the underestimation or misrepresentation of an analog’s effects in clinical settings [40]. Given the marked phylogenetic and metabolic disparities between rodents and humans, the translation of preclinical findings from murine models to clinical efficacy remains a critical challenge in metabolic drug development. To bridge this gap and robustly validate the therapeutic activity of FGF21-164, studies must go further and be conducted on non-human primates, including immune profiling and toxicity kinetics. Second, while our experiments show no immediate safety concerns, the absence of chronic toxicity and immunogenicity data represents a significant gap. Prolonged exposure to FGF21-164 could theoretically induce antibody formation or tissue-specific toxicity, particularly given the heterologous CD164-TR fusion partner. Third, the current dose regimen, though effective in reducing hyperglycemia and steatosis, has not been systematically optimized through dose–response studies. A narrow therapeutic window—common to many FGF21 analogs—may necessitate tailored dosing strategies to balance efficacy with potential off-target effects. Finally, the CD164-TR domain, while engineered to extend half-lives, may interact with unintended molecular targets (e.g., extracellular matrix receptors or immune modulators), potentially confounding the observed metabolic improvements. To address these limitations, future work will prioritize (1) validation in human hepatocyte organoids and non-human primates to bridge species-specific gaps, (2) chronic toxicity studies conducted on primates with immunophenotyping, (3) dose-ranging trials coupled with pharmacokinetic/pharmacodynamic modeling, and (4) proteomic profiling (e.g., affinity pulldown MS) to map CD164-TR interactions. The transparent reporting of these limitations provided here underscores the need for cautious interpretation while guiding subsequent preclinical development.

The FGF21-164 fusion protein developed in this study exhibited long-lasting pharmacokinetic properties and enhanced pharmacological efficacy in mouse models, demonstrating its effect on glycemic and body weight control. Collectively, our findings highlight the utility of the TR fusion carrier of hCD164 in the development of biologics with optimized pharmacological profiles. The improved pharmacokinetic and pharmacodynamic properties of FGF21-164 underscore its therapeutic potential for addressing metabolic disorders, particularly the obesity-associated dysregulation of glucose and lipid metabolism, and nonalcoholic steatohepatitis.

## 4. Materials and Methods

### 4.1. Construction of FGF21-164-Expressing Cell Lines

FGF21-164, an engineered FGF21 analog, was constructed by fusing a human FGF21 variant (ΔHPIP, L118C, and A134C) with a tandem repeat (TR) amino acid sequence (Ser110-Asp162 of hCD164). Both the FGF21-164 and FGF21 genes were codon-optimized for recombinant expression and commercially synthesized (GENEWIZ, Suzhou, China), with their detailed sequences illustrated in Figure 1A. These gene fragments were then cloned into expression vectors (pD2531, ATUM, Newark, CA, USA) and then expressed recombinantly through stable transfection of serum-free adapted 293F cells. Recombinant cell lines lacking glutamine were selected for growth media, and subclones were screened in 96-well tissue culture plates using limiting dilutions.

### 4.2. Identification, Fermentation, and Purification of FGF21-164

Recombinant protein expression was initially characterized using Western blot analysis via an anti-FGF21 antibody (#50421-R005, SinoBiologicals, Beijing, China), followed by protein fermentation and purification. The FGF21-164-expressing cell lines were cultured in 2 L medium (Thermo Fisher, Waltham, MA, USA) for 4 days. Post-harvest, the culture supernatant containing FGF21-164 was centrifuged and filtered. Initial purification was performed using an IexCap DEAE 6FF column (SMART LIFESCIENCES, Changzhou, China); this was followed by pooling and tenfold concentration of recombinant-protein-containing fractions (verified by SDS–PAGE and anti-FGF21 immunoblotting) via ultrafiltration (Millipore, Burlington, MA, USA, 10 kDa MWCO). Further purification was achieved using size exclusion chromatography (HiLoad 16/60 Superdex 75 pg, GE Healthcare, Chicago, IL, USA). FGF21 was purified using an identical protocol.

### 4.3. Size Exclusion Chromatography (SEC) Analysis of FGF21-164

SEC was conducted using an AdvanceBio SEC 300 Å column produced by Agilent Technologies (Santa Clara, CA, USA), with the chromatographic conditions set as described earlier [16]. FGF21-164 (0.5 mg·mL^−1^, 5 µL) was injected into sodium phosphate buffer (150 mmol·L^−1^, pH 7.0), and peak elution times were recorded. The hydrodynamic-radius-derived apparent molecular size was determined via interpolation from a standard calibration curve (R^2^ = 0.999) generated using the protein standards (Agilent Technologies, Santa Clara, CA, USA).

### 4.4. Mass Spectroscopy

Mass-spectrometry analysis was performed using a Bruker Autoflex Speed system, using 2,5-dihydroxybenzoic acid as the matrix. Spectral data processing of FGF21-164 was conducted using the software that comes with the instrument.

### 4.5. Circular Dichroism (CD) Analysis

Far-UV circular dichroism (CD) spectra were recorded using a Chirascan spectropolarimeter (Applied PhotoPhysics Ltd., Leatherhead, Surrey, UK) across 200–260 nm. Measurements were performed in triplicate at room temperature using the standard process [16]. Buffer background subtraction was performed using sodium phosphate reference scans, with data subsequently converted to molar ellipticity units.

### 4.6. Glycan Analysis of FGF21-164

For N-glycan analysis, FGF21-164 was subjected to dissolution, denaturation, and PNGase F digestion, as previously described [16]. We purified and vacuum-dried the released N-glycans. Following 2-aminobenzamide labeling, N-glycans were analyzed using normal-phase UPLC, as previously described [41]. Quantification was performed using maltopentaose as an external standard, employing a linear calibration model established under identical analytical conditions.

O-glycans were released through optimized β-elimination, as previously described [16]. The reaction was terminated via vacuum centrifugation, with water added repeatedly to remove residual ammonium salts. Dried samples were treated with 20 µL of formic acid (1%) in the dark for 40 min at 25 °C. Purification via carbon SPE columns was conducted, followed by 2-AB labeling and UPLC analysis using the same protocol as for N-glycans.

### 4.7. 3T3-L1 Cell Culture and Differentiation

3T3-L1 cells from Wuhan Pricella Biotechnology Co., Ltd. (Wuhan, China), were cultured in DMEM (Cytiva, Marlborough, MA, USA) with 10% calf bovine serum (ExCell Bio, Suzhou, China) until reaching contact inhibition, followed by a 2-day post-confluent phase. Adipocyte differentiation was initiated by switching to DMEM containing 10% FBS (ExCell Bio, Suzhou, China) and an MDI cocktail: 0.5 mM IBMX, 1 μM of dexamethasone (both from Sigma, St. Louis, MO, USA), and 10 μg·mL^−1^ of insulin (Sigma, St. Louis, MO, USA). After 48 h, the medium was replaced with DMEM supplemented with 10% FBS and 10 μg·mL^−1^ of insulin for an additional 2 days. Maintenance was performed by feeding the culture with 10% FBS/DMEM every 48 h until day 10, when differentiated adipocytes displaying characteristic rounded morphology and lipid droplet accumulation were observed.

### 4.8. Glucose Uptake

Differentiated 3T3-L1 adipocytes, prepared as described, were seeded in 48-well plates at 5.0 × 10^4^ cells/well. Following serum starvation in 0.1% FBS/DMEM for 24 h, cells were treated with FGF21-164 (0–100 μg·mL^−1^) in growth medium for 24 h. Glucose uptake was quantified by measuring medium glucose depletion using a commercial assay kit (Nanjing Jiancheng Bioengineering Institute, Nanjing, China) according to the manufacturer’s instructions.

### 4.9. Western Blotting

Differentiated adipocytes were serum-starved in 0.1% FBS/DMEM for 24 h and then treated withFGF21-164 (40 μg·mL^−1^) for various durations (0, 2, 10, 30, 60, and 120 min). The cells were collected and cleaned and then lysed using RIPA buffer (Yeasen Biotech, Shanghai, China) supplemented with phosphatase and protease inhibitor cocktails (Yeasen Biotech, Shanghai, China). The lysates were centrifuged to collect the supernatant, and protein concentrations were determined using Bradford reagent (Sangon Biotech, Shanghai, China).

For phospho-Erk1/2 detection, total proteins were loaded on SDS-PAGE gels. We then separated the specific protein bands and transferred them to PVDF membranes. After being blocked with QuickBlock™ Blocking Buffer (Beyotime Biotech, Beijing, China) for 30 min, the membranes were incubated with primary antibodies: rabbit anti-phospho-Erk1/2 (#4370, Cell Signaling Technology, Danvers, MA, USA) and rabbit anti-Erk1/2 (#4695, Cell Signaling Technology, Danvers, MA, USA). Following TBS-T washes, membranes were probed with secondary antibody (1:5000). After additional washes, protein bands were visualized using an ECL kit (Yeasen Biotech, Shanghai, China).

### 4.10. Measurement of Oil Droplets in Adipocytes via Oil Red O Staining

3T3-L1 cells were plated in 48-well plates at 5.0 × 10^4^ cells/well and differentiated using MDI and insulin. Cells were maintained in 10% FBS/DMEM with or without 40 μg·mL^−1^ of FGF21-164, with medium changes carried out every 48 h until day 10. Three experimental groups were established: FGF21-164-treated differentiated cells (FGF21-164), untreated differentiated cells (Model), and undifferentiated controls (Ctrl).

Lipid accumulation was assessed using Oil Red O staining (Sbjbio, Beijing, China). Cells were PBS-washed, fixed for 30 min at 25 °C, and completely dried. After being stained with working solution, the samples were washed with 60% isopropanol to remove background staining. Lipid droplets were observed and imaged using a light microscope. For quantification, stained cells were extracted with isopropanol (40 min) with 10 min of shaking, and supernatant absorbance was measured at 510 nm.

### 4.11. Animal Experiments

All animal procedures were conducted in compliance with the NIH Guidelines for the Care and Use of Laboratory Animals and approved by the Ethics Committee of China Pharmaceutical University (consent No. 2024-04-014).

Pharmacokinetic evaluation in mice. Twelve 8-week-old male C57BL/6J mice (Qinglongshan Animal Breeding Center, Nanjing, China) were randomly divided into three groups and administered FGF21-164 (22.7 mg·kg^−1^) via intravenous injection. Blood samples were collected through retro-orbital puncture at designated intervals. Serum was separated via centrifugation (11,000 rpm, 4 °C, 5 min) and analyzed using UHPLC-MS/MS. A surrogate peptide (YLYTDDAQQTEAHLEIR) was identified through Skyline software, version 21.1 prediction and high-resolution MS verification. Serum samples were PBS-diluted, heat-denatured (60 °C), alkylated, and trypsin-digested (37 °C, 2 h) for peptide preparation. Chromatographic separation was achieved on an EclipsePlus C18 column with mobile phases: 0.1% formic acid in water (A) and 0.1% formic acid in acetonitrile (B). Quantitative analysis was performed using external calibration with a triple-quadrupole mass spectrometer. Pharmacokinetic parameters were calculated using Phoenix WinNonlin 7.0 (Pharsight, Mountain View, CA, USA) via noncompartmental analysis.

Measurement of blood glucose levels. We used a Roche ACCU-CHEK Performa (Roche, Basel, Switzerland) glucometer, which is an autologous blood glucose meter, to measure blood glucose, using Accu-CHEK Performa test strips as dedicated sensors. Blood was taken from the tail tip for measurement.

Single-dose study. For single-dose evaluation, 8-week-old male *ob*/*ob* mice (B6/JGpt-*Lep^em1Cd25^*/Gpt) were obtained from GemPharmatech (Nanjing, China) and randomly divided into three groups: model (PBS), 6 mg·kg^−1^ of FGF21-164, and 12 mg·kg^−1^ of FGF21-164. Following a 12 h fast, mice were administered subcutaneous injections of respective treatments (200 µL). Blood glucose levels were monitored at 0, 2, 3, 4, 5, 6, 7, and 8 h post-administration.

Oral glucose tolerance test. Oral glucose tolerance tests (OGTTs) were conducted on *ob*/*ob* mice at 7, 21, and 28 days after single-dose FGF21-164 administration. Following a 12 h fast, mice received an oral glucose load (1.5 g·kg^−1^), with blood glucose levels measured at 0–120 min. The areas under the curve (AUCs) were calculated using the trapezoidal method.

Repeated-administration study. Diet-induced obese (DIO) mice, a well-established model for obesity and hepatic steatosis [42], were generated by feeding 6-week-old male C57BL/6J mice (Qinglongshan Animal Breeding Center, Nanjing, China) a high-fat diet (HFD; 60 kcal% fat) for 14 weeks. Mice exceeding 30 g were randomly assigned to the model or FGF21-164 groups, while standard diet (SD)-fed mice served as the Ctrl group. Treatments were administered intraperitoneally every 48 h: PBS (200 µL) for Ctrl and model groups, and FGF21-164 (3 mg·kg^−1^, 200 µL) for the treatment group. Body weights were recorded daily. After 19 days, mice were euthanized for hepatic lipid analysis using Oil Red O staining (Nanjing FreeThinKing Biotechnology Co., Ltd. Nanjing, China).

### 4.12. Statistical Analysis

Data are expressed as means ±SEM. Analyses included one-way ANOVA for multiple comparisons using Tukey’s test and two-way ANOVA for multi-parameter studies using Sídák’s test, with significance set at *p* < 0.05.

## Figures and Tables

**Figure 1 ijms-26-02672-f001:**
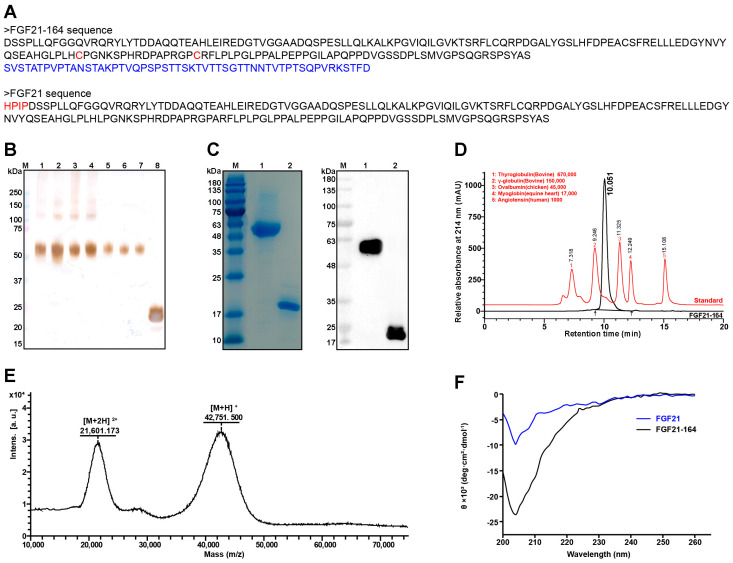
Construction, purification, and characterization of FGF21-164. (**A**) FGF21-164 sequence and FGF21 sequence. In the FGF21-164 sequence, red marked the mutated amino acid and blue represents the TR acid sequence from hCD164. In the FGF21 sequence, the red-labeled amino acid represents the amino acid sequence whose n-terminal is deleted in FGF21-164. (**B**) Subcloned cell culture supernatants expressing FGF21-164 and FGF21 were analyzed using Western blotting (Lane M: molecular weight marker. Lane 1~7: FGF21-164 clones 1, 2, 3, A4, B2, F3, and H4. Line 8: FGF21 clone2). (**C**) The purified FGF21-164 and FGF21 were assayed using SDS-PAGE and Western blotting (Lane M: molecular weight marker. Lane 1: purified FGF21-164. Line 2: purified FGF21). (**D**) Purity and hydrodynamic radius of FGF21-164 were analyzed using SEC-HPLC. (**E**) MALDI-TOF mass-spectrometry analysis of FGF21-164. ([M + H] ^+^ denotes the singly charged ionic forms, and [M + 2H] ^2+^ denotes doubly charged ionic forms). (**F**) Circular Dichroism (CD) spectra of FGF21-164 and FGF21.

**Figure 2 ijms-26-02672-f002:**
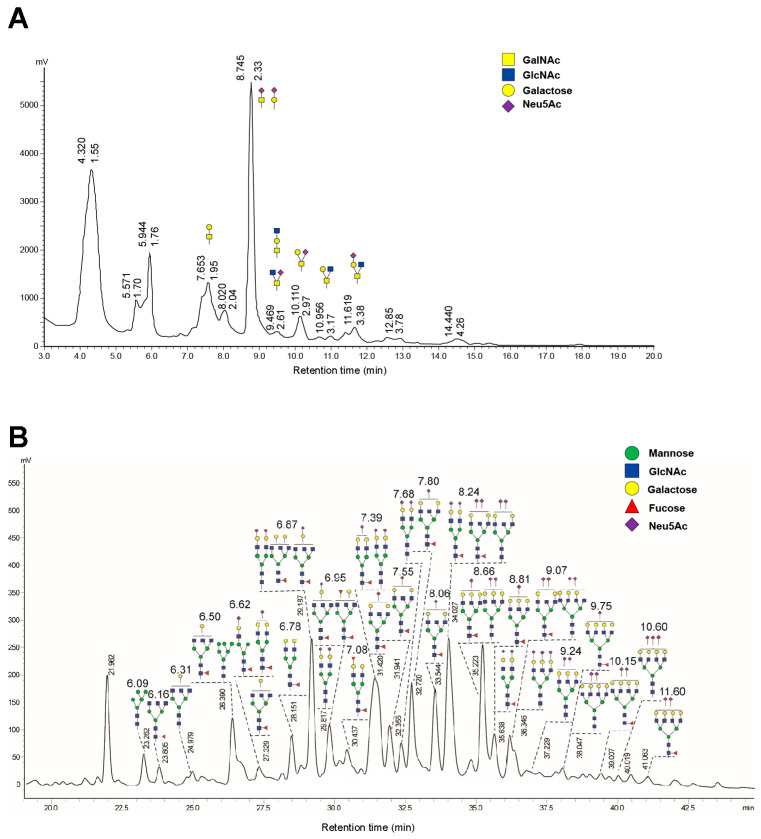
Glycan forms analysis of the protein FGF21-164. (**A**) O-linked glycan forms of FGF21-164. (**B**) N-linked glycan forms of FGF21-164.

**Figure 3 ijms-26-02672-f003:**
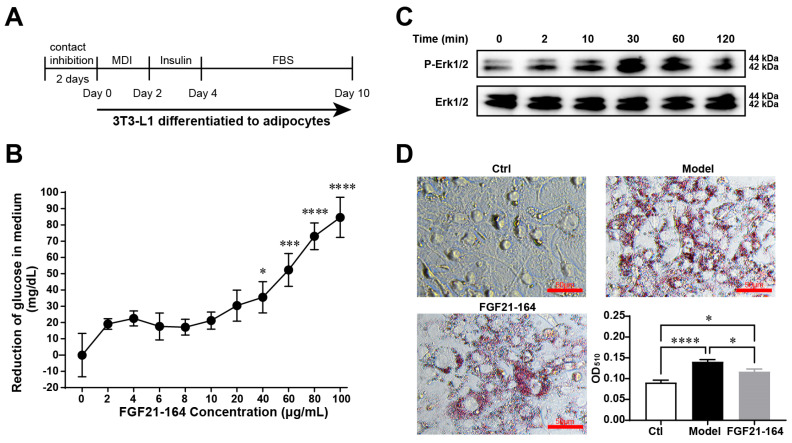
Effect of FGF21-164 in adipocytes induced by 3T3-L1. (**A**) The process of inducing 3T3-L1’s differentiation into adipocytes. (**B**) Reduction in glucose in medium stimulated by FGF21-164 in adipocytes induced by 3T3-L1. Data are means ± SEM (n = 6). * *p* < 0.05, *** *p* < 0.001, and **** *p* < 0.0001 compared to 0 μg/mL FGF21-164. (**C**) Phospho-Erk1/2-specific bands (Thr202/Tyr204; 44–42 kDa) were detected in 3T3-L1-derived adipocytes after FGF21-164 stimulation. (**D**) Evaluation of lipid droplet accumulation in adipocytes induced by 3T3-L1 with or without FGF21-164. The scale bar represents 50 μm.Data are the means ± SEM (n = 6). * *p* < 0.05, **** *p* < 0.0001.

**Figure 4 ijms-26-02672-f004:**
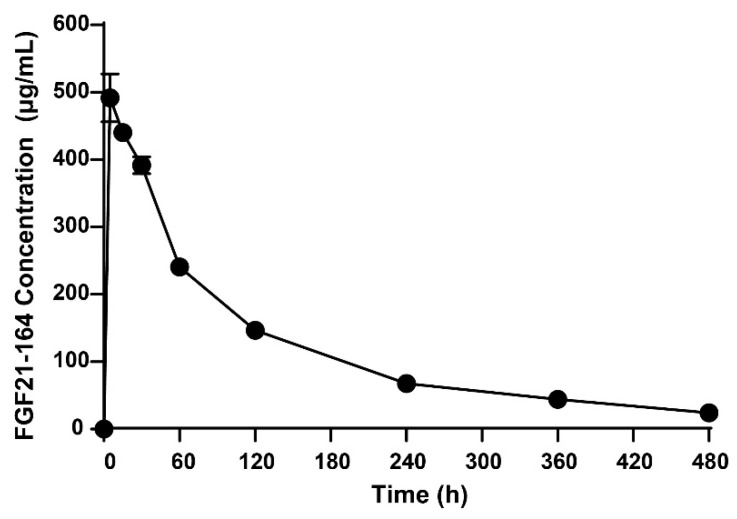
Pharmacokinetics of FGF21-164. FGF21-164 plasma levels in C57BL/6 mice as determined via targeted LC-MS after i.v. (tail vein) bolus injection of FGF21-164 protein.

**Figure 5 ijms-26-02672-f005:**
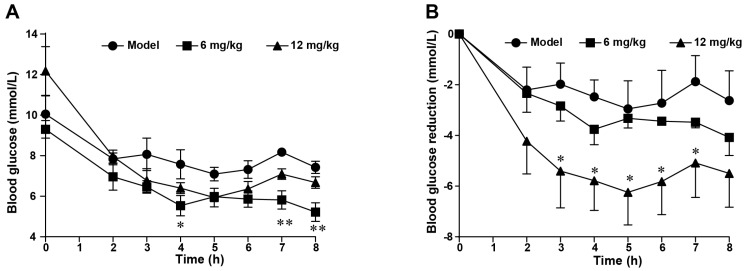
The results of an oral glucose tolerance test applied to *ob*/*ob* mice after a single FGF21-164 treatment. (**A**) Blood glucose levels were measured at 0, 2, 3, 4, 5, 6, 7, and 8 h after the administration of FGF21-164. Data are the means ± SEM (n = 4–5). * *p* < 0.05 and ** *p* < 0.01 compared with the model. (**B**) The decrease in blood glucose levels post-administration relative to the baseline measurement at 0 h. Data are the means ± SEM (n = 4–5). * *p* < 0.05 compared with the model.

**Figure 6 ijms-26-02672-f006:**
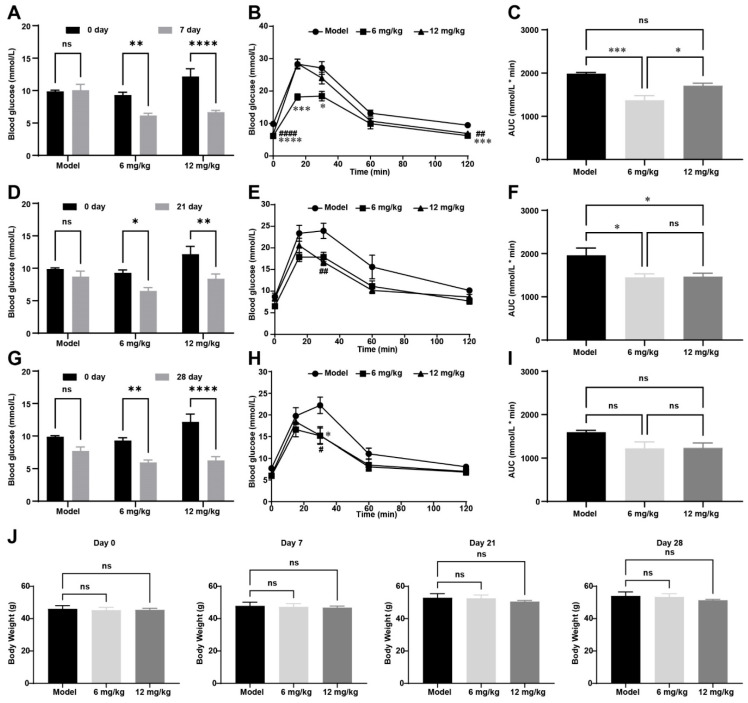
Effect of a single administration of FGF21-164 on *ob*/*ob* mice. (**A**,**D**,**G**) Fasting glucose levels. Blood glucose levels were measured at 7, 21, and 28 days after a single dose of FGF21-164 was administered. Data are the means ± SEM (n = 4–5). ns: no significance. * *p* < 0.05, ** *p* < 0.01, and **** *p* < 0.0001. (**B**,**E**,**H**) The OGTT at 7, 21, and 28 days. After oral glucose administration, blood glucose levels were measured. Data are the means ± SEM (n = 4–5). * *p* < 0.05, *** *p* < 0.001, and **** *p* < 0.0001 at 6 mg·kg^−1^ vs. model. # *p* < 0.05, ## *p* < 0.01 and #### *p* < 0.0001 at 12 mg·kg^−1^ vs. model. (**C**,**F**,**I**) The glucose AUC during the OGTT process. Data are the means ± SEM (n = 4–5). ns: no significance. * *p* < 0.05, *** *p* < 0.001. (**J**) Body weight measured at 0, 7, 21, and 28 days. Data are the means ± SEM (n = 4–5). ns: no significance.

**Figure 7 ijms-26-02672-f007:**
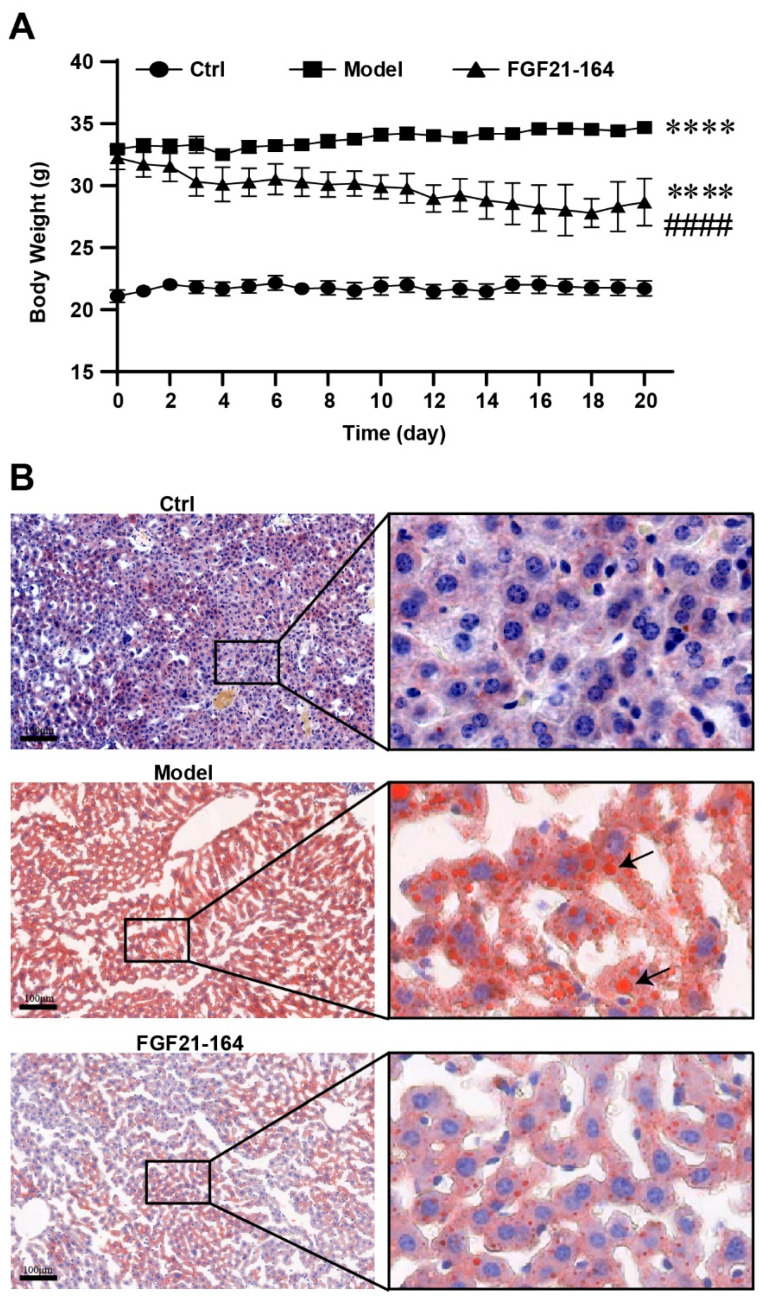
Effect of repeated administration of FGF21-164. (**A**) Body weight changes. The DIO mice were treated with PBS or FGF21-164 once every 2 days. Data are the means ± SEM (n = 5–7). **** *p* < 0.0001 vs. Ctrl. #### *p* < 0.0001 for FGF21-164 vs. the model. (**B**) The liver sections were stained with Oil-red-O. The scale bar represents 100 μm. DIO mice showed many red lipid droplets in hepatic tissue (indicated by the black arrow), while lipid droplet levels were lower after the FGF21-164 treatment.

## Data Availability

The data that support the findings of this study are available from the corresponding author upon reasonable request. Some data may not be made available because of privacy or ethical restrictions.

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
