# Peer review of "A Novel Recombinant Human FGF21 Analog with High Glycosylation Has a Prolonged Half-Life and Affects Glycemic and Body Weight Control"

_ijms, 2025, doi:10.3390/ijms26062672_

Round 1

Reviewer 1 Report

Comments and Suggestions for Authors

Reviewer Comments

  1. Therefore, FGF21-164 in ob/ob mice may reduce appetite and influence body weight across the entire population. Is there any interaction between FGF21-164 and body weight?
  2. Conclusion: The final sentence should be relocated to the results section, while its implications can be incorporated into the conclusion section, similar to how they are addressed in the manuscript's conclusion.
  3. Line 71-76: The final sentence in the methodology section should be relocated to a more appropriate place, as it describes the specific roles of certain authors.
  4. Line 406: This section is too short. I suggest that the authors revise lines.
  5. It is necessary to ensure consistency in the decimal points for the p-values.
  6. The negative correlation observed between glycemic control and weight reduction in the univariate analysis is not surprising.
Comments on the Quality of English Language

Reviewer Comments

  1. Therefore, FGF21-164 in ob/ob mice may reduce appetite and influence body weight across the entire population. Is there any interaction between FGF21-164 and body weight?
  2. Conclusion: The final sentence should be relocated to the results section, while its implications can be incorporated into the conclusion section, similar to how they are addressed in the manuscript's conclusion.
  3. Line 71-76: The final sentence in the methodology section should be relocated to a more appropriate place, as it describes the specific roles of certain authors.
  4. Line 406: This section is too short. I suggest that the authors revise lines.
  5. It is necessary to ensure consistency in the decimal points for the p-values.
  6. The negative correlation observed between glycemic control and weight reduction in the univariate analysis is not surprising.

Author Response

Reviewer Comments

1. Therefore, FGF21-164 in ob/ob mice may reduce appetite and influence body weight across the entire population. Is there any interaction between FGF21-164 and body weight?

We sincerely thank the reviewer for raising this important question regarding the potential interaction between FGF21-164 and body weight.

Our observations in ob/ob mice are consistent with the known physiological effects of FGF21-164. In ob/ob mice treated with FGF21-164, we observedthe magnitude of weight loss was not as significant as the improvement in blood sugar control (Figure 6J). FGF21 analogs are known to improve insulin sensitivity and reduce hyperglycemia independent of weight loss in certain contexts. Leptin involvement was suggested by the inability of FGF21-class molecules to effectively induce weight loss in ob/ob or db/db mice that lack intact leptin signaling [1, 2, 3].

Importantly, in diet-induced obese (DIO) mice, a model of obesity with intact leptin signaling, FGF21-164 repeated administration induced significant weight loss (Figure 7A). FGF21 are known to influence hypothalamic pathways that modulate appetite and energy expenditure. While our study did not directly measure food intake or energy expenditure, the observed weight reduction in DIO mice may reflect a combination of reduced caloric intake (via central signaling) and increased lipid oxidation (via peripheral actions in adipose tissue and liver). These mechanisms are consistent with prior studies on native FGF21 and its analogs [4, 5, 6].

We acknowledge that the current study did not include detailed analyses of appetite-regulating hormones (e.g., leptin, ghrelin) or indirect calorimetry to dissect the contributions of energy intake versus expenditure. These aspects will be prioritized in follow-up studies to fully elucidate the mechanistic interplay between FGF21-164, body weight, and metabolic outcomes. This part was highlighted in the discussion on Line 297-303.

2. Conclusion: The final sentence should be relocated to the results section, while its implications can be incorporated into the conclusion section, similar to how they are addressed in the manuscript's conclusion.

We sincerely thank the reviewer for this constructive suggestion to improve the manuscript’s organization. In response to the comment, we rewrote the Conclusion on Line 333-340. This adjustment ensures that specific data-driven statements remain within the context of results presentation.

In the revised Conclusion section, we have reframed the implications of this finding to focus on its broader significance, aligning with the study’s translational goals. The FGF21-164 fusion protein developed in this study exhibited prolonged phar-macokinetic properties and enhanced pharmacological efficacy in mouse models, demonstrating its effect on glycemic and body weight control. Collectively, our findings highlight the utility of the TR fusion carrier of hCD164 in the development of biologics with optimized pharmacological profiles. The improved pharmacokinetic and pharmacodynamic properties of FGF21-164 underscore its therapeutic potential for ad-dressing metabolic disorders, particularly obesity-associated dysregulation of glucose and lipid metabolism, and nonalcoholic steatohepatitis.

This reorganization enhances the logical flow of the manuscript, ensuring that the Conclusion section synthesizes overarching insights rather than reiterating specific results. We appreciate the reviewer’s guidance in refining the manuscript’s structure and clarity.

3. Line 71-76: The final sentence in the methodology section should be relocated to a more appropriate place, as it describes the specific roles of certain authors.

Thanks for the suggestion. In the revised manuscript, we described this methodology on Line 352-355.

4.2. Identification, fermentation and purification of FGF21-164

Recombinant protein expression was characterized initially via Western blot analysis via an anti-FGF21 antibody (#50421-R005, SinoBiologicals, Beijing, China), followed by protein fermented and purified.

4. Line 406: This section is too short. I suggest that the authors revise lines.

Thanks for the suggestion. In the revised manuscript, we explain in detail in Line 234-247 how sialic acid prevents protein degradation in the liver.

These sialylation modifications confer FGF21-164 a key advantage in extending its half-life. First, sialylation-mediated masking of galactose residues in FGF21-164 effectively inhibits recognition by the asialoglycoprotein receptor (ASGPR) abundantly expressed on hepatic sinusoidal endothelial cells. The negatively charged carboxyl groups of sialic acids create steric and electrostatic repulsion, further inhibiting ASGPR binding. Second, Sialylation increases FGF21-164 hydrophilicity, reducing aggregation propensity and improving plasma solubility. Concurrently, the negative surface charge repels similarly charged basement membranes and podocytic epithelia in the glomerulus, synergistically decreasing renal clearance rates. Additionally, sialic acid capping may mitigate immune recognition of FGF21-164, thereby reducing phagocytic uptake and proteolytic degradation.

5. It is necessary to ensure consistency in the decimal points for the p-values.

Thank you for your comment. We agree that maintaining consistency in the decimal points for p-values is important for clarity and precision. In the revised manuscript, we have standardized the presentation of p-values to ensure uniformity throughout the text, figures, and tables. We appreciate your attention to this detail.

6. The negative correlation observed between glycemic control and weight reduction in the univariate analysis is not surprising.

We appreciate the reviewers' views.

FGF21 analogs are known to improve insulin sensitivity and reduce hyperglycemia independent of weight loss in certain contexts. Our experimental data demonstrated that FGF21-164 administration ameliorated hyperglycemia in ob/ob mice, and improved glucose tolerance (Figure 6). However, no statistically significant alterations in body weight were observed between treated and vehicle groups (Figure 6J). Leptin involvement was suggested by the inability of FGF21-class molecules to effectively induce weight loss in ob/ob or db/db mice that lack intact leptin signaling [1, 2, 3].

Reference:

[1] Kolumam, G., Chen, M. Z., Tong, R., Zavala-Solorio, J., Kates, L., van Bruggen, N., Ross, J., Wyatt, S. K., Gandham, V. D., Carano, R. A., Dunshee, D. R., Wu, A. L., Haley, B., Anderson, K., Warming, S., Rairdan, X. Y., Lewin-Koh, N., Zhang, Y., Gutierrez, J., Baruch, A., … Sonoda, J. (2015). Sustained Brown Fat Stimulation and Insulin Sensitization by a Humanized Bispecific Antibody Agonist for Fibroblast Growth Factor Receptor 1/βKlotho Complex. EBioMedicine, 2(7), 730–743. https://doi.org/10.1016/j.ebiom.2015.05.028

[2] Wu, A. L., Coulter, S., Liddle, C., Wong, A., Eastham-Anderson, J., French, D. M., Peterson, A. S., & Sonoda, J. (2011). FGF19 regulates cell proliferation, glucose and bile acid metabolism via FGFR4-dependent and independent pathways. PloS one, 6(3), e17868. https://doi.org/10.1371/journal.pone.0017868

[3] Berglund, E. D., Li, C. Y., Bina, H. A., Lynes, S. E., Michael, M. D., Shanafelt, A. B., Kharitonenkov, A., & Wasserman, D. H. (2009). Fibroblast growth factor 21 controls glycemia via regulation of hepatic glucose flux and insulin sensitivity. Endocrinology, 150(9), 4084–4093. https://doi.org/10.1210/en.2009-0221

[4] Coskun, T., Bina, H. A., Schneider, M. A., Dunbar, J. D., Hu, C. C., Chen, Y., Moller, D. E., & Kharitonenkov, A. (2008). Fibroblast growth factor 21 corrects obesity in mice. Endocrinology, 149(12), 6018–6027. https://doi.org/10.1210/en.2008-0816

[5] Xu, J., Lloyd, D. J., Hale, C., Stanislaus, S., Chen, M., Sivits, G., Vonderfecht, S., Hecht, R., Li, Y. S., Lindberg, R. A., Chen, J. L., Jung, D. Y., Zhang, Z., Ko, H. J., Kim, J. K., & Véniant, M. M. (2009). Fibroblast growth factor 21 reverses hepatic steatosis, increases energy expenditure, and improves insulin sensitivity in diet-induced obese mice. Diabetes, 58(1), 250–259. https://doi.org/10.2337/db08-0392

[6] Kolumam, G., Chen, M. Z., Tong, R., Zavala-Solorio, J., Kates, L., van Bruggen, N., Ross, J., Wyatt, S. K., Gandham, V. D., Carano, R. A., Dunshee, D. R., Wu, A. L., Haley, B., Anderson, K., Warming, S., Rairdan, X. Y., Lewin-Koh, N., Zhang, Y., Gutierrez, J., Baruch, A., … Sonoda, J. (2015). Sustained Brown Fat Stimulation and Insulin Sensitization by a Humanized Bispecific Antibody Agonist for Fibroblast Growth Factor Receptor 1/βKlotho Complex. EBioMedicine, 2(7), 730–743. https://doi.org/10.1016/j.ebiom.2015.05.028

Reviewer 2 Report

Comments and Suggestions for Authors

The manuscript by Pei Du et al. provides an in-depth analysis of the role of FGF21 analogue as a hormone-like peptide and its involvement in blood glucose and lipid metabolism.

The introduction is well written and the references are adequately cited; however, the hypothesis of the study is not fully recognisable in this form (it would be preferable to rephrase this hypothesis in order to facilitate understanding for the reader).Although the manuscript contains all the standard sections for this type of journal, their position in the text does not meet the requirements of the IJMS.The research findings and conclusions provide a solid basis for further studies.

The planned studies provide substantial evidence to support the research hypothesis, demonstrating a clear and strong fit between the experimental design and the proposed aims.The results are presented in a clear and unambiguous manner, with all necessary information provided in a reliable and understandable manner, thus facilitating the reader's ability to understand the study's conclusions.

The discussion section, however, is characterised by a lack of organisation and clarity, which hinders the identification of the consistency or contradiction of the data obtained with the existing literature on the subject.The lack of coherence in the presentation further hinders the reader's ability to fully grasp the significance of the findings and their place within the wider context of current research.Furthermore, the discussion section lacks a crucial element: a dedicated section on the limitations of the study. It is imperative that the limitations of the study are addressed in order to provide a balanced and transparent assessment of the research, offering a more nuanced interpretation of the findings and highlighting areas for further investigation.

Details and questions :

  • Firstly, it is evident that additional information regarding protein size should be incorporated into Figure 3C, and the figure description should be refined.
  • Secondly, in Figure 7, the IHC images should be of increased size and quality, with arrows that accurately indicate the most significant findings.
  • Thirdly, the authors should confirm whether they observed any (histo-)pathological changes that are characteristic of NAFLD.
  • Despite the article's focus on the impact of FGF21-164 on glucose and lipid metabolism, the molecular mechanisms underlying its efficacy and safety remain unaddressed, a crucial aspect for evaluating the therapeutic potential. While the study highlights the beneficial effects of FGF21-164, it lacks a comprehensive analysis of potential side effects, including immunological reactions and toxicity. It is unclear whether the authors have endeavoured to assess these factors.
  • The Acknowledgements section discloses that Jiangsu Cell Tech Medical Research Institute Co., Ltd has filed a patent, which may indicate a lack of objectivity in the research findings.
  • The authors should have analysed the long-term effects of the FGF21-164 analogue.
  • Extending these studies with spatial transcriptomics analyses would be beneficial, as these would show transcriptome changes with a histological approach.
  • Furthermore, the translational nature of the research conducted is questionable, as the authors focus on an animal model, which does not always correspond to human tissues.

Author Response

Comments and Suggestions for Authors

1. The manuscript by Pei Du et al. provides an in-depth analysis of the role of FGF21 analogue as a hormone-like peptide and its involvement in blood glucose and lipid metabolism.

The introduction is well written and the references are adequately cited; however, the hypothesis of the study is not fully recognisable in this form (it would be preferable to rephrase this hypothesis in order to facilitate understanding for the reader).Although the manuscript contains all the standard sections for this type of journal, their position in the text does not meet the requirements of the IJMS. The research findings and conclusions provide a solid basis for further studies.

Thanks for the suggestion. In the revised manuscript, we rewrote this section on Line 56-61.

Here, we describe a novel human FGF21 analog, designated as FGF21-164, which is generated through fusion protein technology and co-encoded by a mutated FGF21 gene and the TR gene of hCD164. FGF21-164 was expressed via a eukaryotic cell, and its physicochemical and pharmacokinetic properties were thoroughly evaluated. Subsequently, the biological activity of FGF21-164 was assessed at the cellular level. Further-more, FGF21-164 was tested in ob/ob and diet-induced obese (DIO) mice to demonstrate its activity.

In addition, in accordance with the requirements of the IJMS journal, we have reorganized the sections of the manuscript to enhance readability and facilitate a better understanding for the readers.

2. The planned studies provide substantial evidence to support the research hypothesis, demonstrating a clear and strong fit between the experimental design and the proposed aims. The results are presented in a clear and unambiguous manner, with all necessary information provided in a reliable and understandable manner, thus facilitating the reader's ability to understand the study's conclusions.

The discussion section, however, is characterised by a lack of organisation and clarity, which hinders the identification of the consistency or contradiction of the data obtained with the existing literature on the subject. The lack of coherence in the presentation further hinders the reader's ability to fully grasp the significance of the findings and their place within the wider context of current research. Furthermore, the discussion section lacks a crucial element: a dedicated section on the limitations of the study. It is imperative that the limitations of the study are addressed in order to provide a balanced and transparent assessment of the research, offering a more nuanced interpretation of the findings and highlighting areas for further investigation.

We sincerely thank the reviewer for their critical feedback on the organization and clarity of the Discussion section. We acknowledge that the original version lacked sufficient structure and failed to contextualize the findings within the broader literature, which limited the interpretative value of the work. In response to these concerns, we have undertaken the following revisions to improve coherence, transparency, and scholarly rigor.

The reasons for the extended half-life of FGF21-164 are discussed in detail on Line 229-249

Links our findings (e.g., Erk phosphorylation) to known FGF21 signaling pathways and contrasts them with prior studies on native FGF21 (Line 250-253).

Highlights consistencies (glucose-lowering effects) and superiority (The sugar control effect lasted for 28 days after a single treatment) with published literature, emphasizing the novel contributions of our fusion strategy. Line 260-271.

A dedicated subsection has been added to provide a balanced critique of the work on Line 304-332. Including the reliance on murine models, which may not fully recapitulate human metabolic pathophysiology. The absence of chronic toxicity and immunogenicity data, which are critical for clinical translation. The need for dose-response studies to optimize therapeutic windows. Potential off-target effects of the CD164-TR fusion partner, requiring further investigation.

Details and questions:

1. Firstly, it is evident that additional information regarding protein size should be incorporated into Figure 3C, and the figure description should be refined.

Thanks for the suggestion. In the revised manuscript, we added the information about the size of the Erk protein in Figure 3C on Line 126 and Line 127.

Erk1/2 (extracellular signal-regulated kinase, 42-44 kDa) phosphorylation is a critical step in the activation of downstream signaling pathways by FGF21 through its receptor FGFR1 and co-receptor β-Klotho. Changes in Erk1/2 phosphorylation levels directly reflect the intensity of signal transduction following FGF21 binding to its receptors, making it a commonly used indicator for assessing the biological activity of FGF21.

Figure 3. Effect of FGF21-164 in adipocytes induced by 3T3-L1. A: The process of inducing 3T3-L1 to differentiate into adipocytes. B: Reduction of glucose in medium stimulated by FGF21-164 in adipocytes induced by 3T3-L1. Data are the mean ± SEM (n = 6). *P < 0.05, ***P < 0.001, ****P < 0.0001 compared to 0 μg/mL FGF21-164. C: Phospho-ERK1/2-specific bands (Thr202/Tyr204; 44-42 kDa) was detected in 3T3-L1-derived adipocytes after FGF21-164 stimulation. D: Evaluation of lipid droplet accumulation in adipocytes induced by 3T3-L1 with or without FGF21-164. Data are the mean ± SEM (n = 6). *P < 0.05, ****P < 0.0001.

2. Secondly, in Figure 7, the IHC images should be of increased size and quality, with arrows that accurately indicate the most significant findings.

Thanks for the suggestion. In the revised manuscript, we have revised the immunohistochemistry (IHC) images accordingly in Figure 7 on Line 203. The images have been resized to improve clarity, and the resolution has been enhanced to ensure higher quality. Additionally, arrows have been carefully added to precisely highlight the most significant findings, as recommended. DIO mice showed many red lipid droplets in hepatic tissue (indicated by the black arrow). These modifications aim to facilitate a more accurate and comprehensive interpretation of the results.

Figure 7. Effect of repeated administration of FGF21-164. A: Body weight changes. The DIO mice were treated with PBS or FGF21-164 once every 2 days. Data are the mean ± SEM (n = 5–7). ****P < 0.0001 vs. Ctrl. ####P < 0.0001 is FGF21-164 vs. Model. B: The liver sections were stained with Oil-red-O. The scale bar represents 100 μm. DIO mice showed many red lipid droplets in hepatic tissue (indicated by the black arrow), while lipid droplets were reduced after FGF21-164 treatment.

3. Thirdly, the authors should confirm whether they observed any (histo-)pathological changes that are characteristic of NAFLD.

We sincerely thank the reviewer for raising this critical question regarding the histopathological evaluation of NAFLD-related changes in our study. In response to the comment, we clarify the following:

In the revised manuscript, we have included detailed histopathological assessments of liver tissues from DIO mice treated with FGF21-164 (Line 197-202). The image revealed that hepatocytes in the DIO miceexhibited marked cellular hypertrophy (characterized by increased cytoplasmic volume and nuclear enlargement), consistent with lipid accumulation and metabolic stress. In contrast, hepatocytes from the FGF21-164-treated group displayed reduced cross-sectional area compared to the Model, indicative of attenuated steatosis and restored cellular architecture.

While our current analysis focused on steatosis, hallmarks of NAFLD progression, we acknowledge that advanced stages of NAFLD/NASH (e.g., ballooning degeneration, fibrosis) may require longer-term studies or alternative models. We have added a discussion in Line 290-297, emphasizing the need for future work in NASH-specific models (e.g., STAM mice or methionine-choline-deficient diet-fed rodents) to evaluate the analog’s efficacy in reversing fibrosis.

We appreciate the reviewer’s emphasis on this aspect and believe the added data strengthen the translational relevance of FGF21-164 as a potential therapeutic candidate for metabolic liver diseases.

4. Despite the article's focus on the impact of FGF21-164 on glucose and lipid metabolism, the molecular mechanisms underlying its efficacy and safety remain unaddressed, a crucial aspect for evaluating the therapeutic potential. While the study highlights the beneficial effects of FGF21-164, it lacks a comprehensive analysis of potential side effects, including immunological reactions and toxicity. It is unclear whether the authors have endeavoured to assess these factors.

We sincerely appreciate the reviewer’s thoughtful critique regarding the mechanistic and safety evaluations of FGF21-164. We acknowledge that elucidating the molecular mechanisms and comprehensively assessing safety profiles are critical for advancing therapeutic development. Below, we address these points in detail:

In this study we did not observe lethal toxicity. However, we recognize that chronic toxicity and immunogenicity require systematic investigation. To address this, we have added a dedicated subsection in the Discussion (Line 304-318) outlining the studies that need to be further conducted to evaluate repeat dosing in non-human primates, including immune profiling and toxicity kinetics. These results will be critical for IND-enabling studies and will be published separately.

This work was designed as a proof-of-concept study to validate FGF21-164’s therapeutic potential, with mechanistic and chronic safety analyses planned as subsequent phases. We have revised the manuscript to clarify this rationale and emphasize that further mechanistic and safety studies are imperative before clinical translation.

We thank the reviewer for emphasizing these gaps and have revised the text to better contextualize the current findings while transparently acknowledging remaining limitations. Your feedback has significantly strengthened the manuscript’s rigor and transparency.

5. The Acknowledgements section discloses that Jiangsu Cell Tech Medical Research Institute Co., Ltd has filed a patent, which may indicate a lack of objectivity in the research findings.

We acknowledge the reviewer's concern regarding the potential perception of bias due to the disclosed patent filed by Jiangsu Cell Tech Medical Research Institute Co., Ltd. However, we would like to emphasize that this research was conducted in strict adherence to principles of scientific integrity and objectivity. All experiments, data collection, and analyses were performed independently and rigorously, without influence from any external interests. The disclosure of the patent is intended to ensure transparency regarding intellectual property related to this study. The scope of the patent covers the use of the CD164 tandem repeat region as a fusion partner to extend the half-life of therapeutic proteins, and the FGF21-164 used in this study serves merely as an example to demonstrate the half-life-extending capability of the CD164 tandem repeat region. We remain committed to upholding the highest standards of research ethics and impartiality in our work.

6. The authors should have analysed the long-term effects of the FGF21-164 analogue.

We sincerely thank the reviewer for raising this important point regarding the long-term effects of the FGF21-164 analog. While the current study primarily focused on characterizing the acute pharmacological properties, pharmacokinetics, and short-term therapeutic efficacy of FGF21-164 in preclinical models, we fully acknowledge the critical importance of evaluating its long-term safety and sustained metabolic benefits.

In the revised manuscript, we have expanded the Discussion section (Line 316-319, Line 323-326) to explicitly highlight the need for long-term studies. Additionally, we have included a statement in the Conclusions emphasizing that further investigations into therapeutic window profiles are essential before clinical translation (Line 320-323).

We appreciate the reviewer’s constructive feedback and agree that these extended evaluations will significantly strengthen the translational relevance of FGF21-164 as a potential therapeutic agent.

7. Extending these studies with spatial transcriptomics analyses would be beneficial, as these would show transcriptome changes with a histological approach.

We sincerely thank the reviewer for their insightful suggestion to incorporate spatial transcriptomics analyses into our study. We fully agree that integrating transcriptome-wide data with histological spatial resolution would provide valuable insights into tissue-specific molecular changes induced by FGF21-164, particularly in metabolic organs such as liver or adipose tissue. Such an approach could further elucidate the relationship between cellular localization and functional outcomes of the analog.

While the current study focused on establishing the pharmacological profile and therapeutic efficacy of FGF21-164 in preclinical models, we acknowledge that spatial transcriptomics represents a powerful complementary tool to deepen mechanistic understanding. At the time of these experiments, restrictions on research funding limited our ability to implement this methodology. However, we have now initiated collaborations to apply spatial transcriptomics in follow-up investigations, specifically to map the heterogeneity of target engagement and downstream signaling pathways in key metabolic tissues. These future studies will be designed to address the reviewer’s valuable point and strengthen the translational relevance of our findings.

We appreciate this constructive feedback and have added a brief discussion in the revised manuscript (Line 297-303) highlighting the potential of spatial transcriptomics as a next-step approach to refine our understanding of FGF21-164’s mode of action.

8. Furthermore, the translational nature of the research conducted is questionable, as the authors focus on an animal model, which does not always correspond to human tissues.

We sincerely appreciate the reviewer’s critical comment regarding the translational relevance of our study. We acknowledge the inherent limitations of animal models in fully replicating human tissue responses. However, our use of ob/ob and diet-induced obese (DIO) mouse models was carefully selected based on their well-established roles in metabolic research, particularly for evaluating glucose homeostasis and lipid metabolism—key pathways targeted by FGF21 analogs. These models are widely recognized for their predictive value in preclinical studies of metabolic therapeutics.

To address translational concerns, we have included discussions in the manuscript (Line 305-316) highlighting the potential differences between murine and human systems, as well as the need for future validation in non-human primates. Additionally, we emphasize that this work represents a foundational step in characterizing the novel FGF21-164 analog, with a focus on mechanistic insights and proof-of-concept efficacy, which are critical before advancing to human clinical trials. Further studies involving human primary hepatocytes or patient-derived organoids are planned as part of our ongoing research to strengthen translational relevance (Line 326-332).

We thank the reviewer for raising this important point and have revised the text to more explicitly address these considerations.
